# Pore Microstructure and Multifractal Characterization of Lacustrine Oil-Prone Shale Using High-Resolution SEM: A Case Sample from Natural Qingshankou Shale

Shansi Tian [1,2,3], Yuanling Guo [1,2], Zhentao Dong [4,*] and Zhaolong Li [5]

[1] State Key Laboratory of Shale Oil and Gas Enrichment Mechanisms and Effective Development, Beijing 100101, China
[2] Sinopec Key Laboratory of Shale Oil/Gas Exploration and Production Technology, Beijing 100101, China
[3] Institute of Unconventional Oil & Gas, Northeast Petroleum University, Daqing 163318, China
[4] School of Geosciences, China University of Petroleum (East China), Qingdao 266580, China
[5] No. 1 Geological Logging Company of Daqing Drilling Engineering Company, Daqing 163318, China
* Correspondence: b20010024@s.upc.edu.cn

**Abstract:** Pore structure is one of the important parameters for evaluating reservoirs, critical in controlling the storage capacity and transportation properties of hydrocarbons. The conventional pore characterization method cannot fully reflect the pore network morphology. The edge-threshold automatic processing method is applied to extract and quantify pore structures in shale scanning electron microscope (SEM) images. In this manuscript, a natural lacustrine oil-prone shale in the Qingshankou Formation of Songliao Basin is used as the research object. Based on FE-SEM, a high-resolution cross-section of shale was obtained to analyze the microstructure of pores and characterize the heterogeneity of pores by multifractal theory. The stringent representative elementary area (REA) of the SEM cross-section was determined to be $35 \times 35$. Four pore types were found and analyzed in the stringent REA: organic pores, organic cracks, inorganic pores, inorganic cracks. The results showed that inorganic pores and cracks were the main pore types and accounted for 87.8% of the total pore area, and organic cracks were of the least importance in the Qingshankou shale. Inorganic pores were characterized as the simplest pore morphologies, with the largest average MinFeret diameter, and the least heterogeneity. Moreover, the inorganic cracks had a long extension distance and stronger homogeneity, which could effectively connect the inorganic pores. Organic pores were found to be the most complex for pore structure, with the least average MinFeret diameter, but the largest heterogeneity. In addition, the extension distance of the organic cracks was short and could not effectively connect the organic pore. We concluded that inorganic pores and cracks are a key factor in the storage and seepage capacity of the Qingshankou shale. Organic pores and cracks provide limited storage space.

**Keywords:** lacustrine shale; automatic image processing; organic pores; scanning electron microscope; multifractal





## 1. Introduction

Shale oil is stored in organic-rich shale strata with predominantly nano-scale pores, and is an important future alternative energy source for conventional oil and gas in the world [1,2]. The economic exploitation of shale oil is related to its reservoir characteristics, viz. the size and distribution of pore-throats [3], the characteristics of mineral and organic matter composition [4], and the reservoir space characteristics of different material compositions [5]. The complication of lacustrine shale oil exploitation in China is related to the insufficient understanding of the shale pore structure [6–8]. The multi-scale pore-throat system developed in oil-prone lacustrine shales is characterized by diverse pore types [9,10], development of nanoscale pores and strong heterogeneity [11–14]. Therefore, multi-scale and heterogeneous shale pore characterization is a challenging and critical study.



Currently, the most widespread conventional pore characterization methods are performed by mercury injection capillary porosimetry (MICP) [15,16], gas adsorption porosimetry [17,18], and nuclear magnetic resonance (NMR) [19–21]. These methods can only detect connected pores and cannot fully reflect the pore network [22–24]. Moreover, since these methods simplify the pore morphology, the pore heterogeneous information about types, morphologies, mineralogy, and microstructures are not accessible [25,26]. Nevertheless, SEM technique enables in-depth analysis of the microscopic pore structure of shale [27–29]. The heterogeneity of pore types, pore size distribution, and morphological microstructure can be revealed by high-resolution SEM images [22,23,29]. The reliability of pore network characterization depends on the pore extraction accuracy of SEM images. Nowadays, the mainstream pore extraction methods include manual drawing methods, thresholding methods [30], edge detection methods [31], and watershed methods [32]. However, these methods have obvious defects: reliance on geological experience, huge workload, and being ineffective for organic pore identification. The edge-threshold automatic processing (ETAP) method reported here is used to address these above identified issues [29].

The pore size distribution, morphology, and fracture extension in shale have self-similar characteristics, thus fractal theory, especially fractal dimension, is widely used to quantitatively characterize the heterogeneity of pore structure [33–35]. Based on the power law of fractal theory, Wang [36] used nitrogen adsorption experiments to discover the effect of fractal characteristics on the pore structure of shales. In addition, the fractal dimension of shale with different pore structure types has been studied based on MICP [37], NMR or SEM [38], with values between two and three [20,21]. However, some of the complex shale pore networks are not mathematically but statistically similar, and their fractal dimensions change with the observation scale, thus requiring description by multiple fractals or continuous spectra of dimensions, termed multiple fractal spectra [38–40]. At present, there are not many studies that apply multifractal theory to describe the results of shale pore networks.

In this paper, the lacustrine oil-prone shale in the Qingshankou, Songliao Basin was used as the research object. High-resolution SEM images of shale were obtained to analyze the microscopic pore structure heterogeneity. The multifractal methods were used to assess the heterogeneity of shale pore networks. Eventually, the effect of heterogeneity of microscopic pore networks on hydrocarbon storage and transport capacity was discussed.

*Geological Setting*

Songliao Basin [41,42] is located in Northeast China and surrounded by hills and mountains (Figure 1a), which is the most abundant non-marine sedimentary basin with oil and gas resources in the world [41,43]. The oil and gas in the basin generally have a distribution characteristic of upper oil and lower gas [44]. Natural gas is mainly developed in the Huoshiling formation, Shahezi Formation (shale gas) and Yingcheng Formation, during fault depression stage, which are continental coal-bearing pyroclastic rock formations with a maximum thickness of 8000 m [45] (Figure 1b). The oil is mainly developed in the Denglouku formation, Quantou Formation, Qingshankou Formation (shale oil), Yaojia Formation, Nenjiang Formation, Sifangtai Formation and Mingshui Formation, during the depression stage, which are continental clastic oil shale formations with a maximum thickness of about 5000 m [46].

Qingshankou Formation is the main oil source layer (Figure 1b), which can be divided into qn1 member and qn23 member from bottom to top. Organic rich shale in qn1 member is the main oil source rock, with high organic matter abundance. Ro in the main depression region is 1.0–1.6% [42]. It is a key area for shale oil exploration, and it also provided the shale sample for this study.

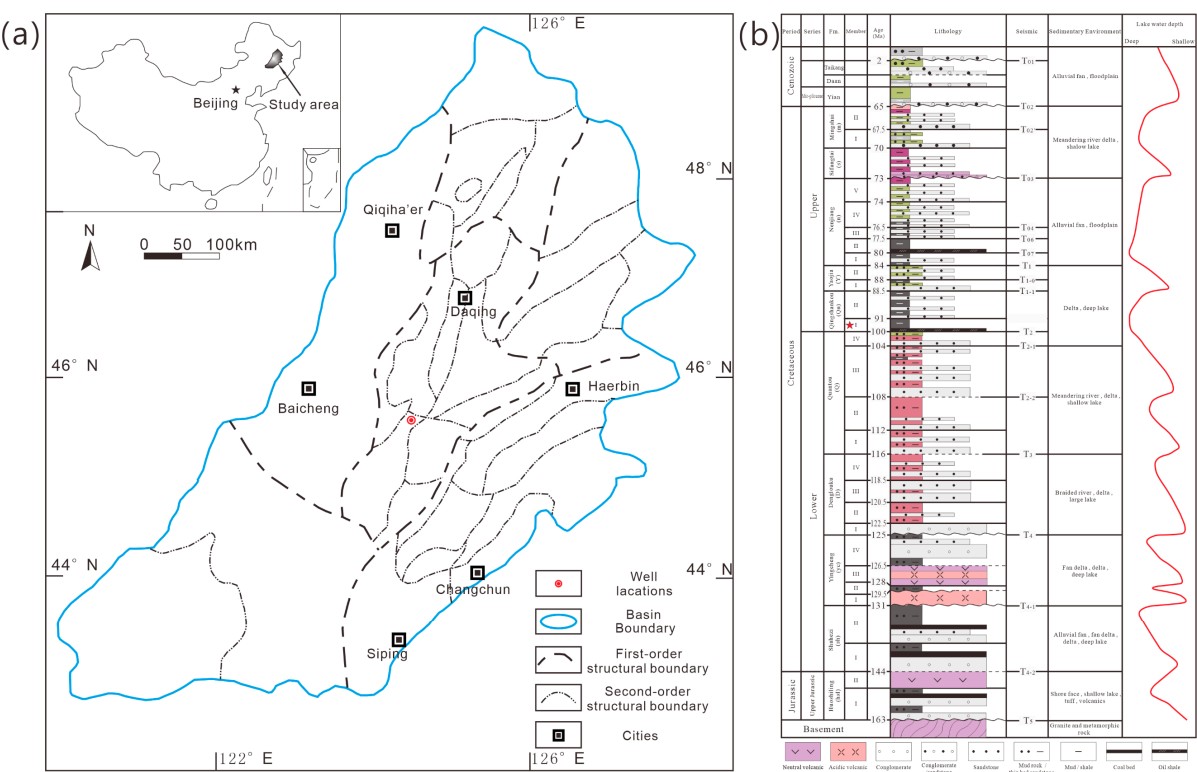

**Figure 1.** (**a**) Sampling position (well X1) and the Songliao Basin. (**b**) Lithological sketch in the study area.

## 2. Materials and Methods

### 2.1. Investigated Samples

In this paper, lamellar oil-prone shale was taken to study the different structure characteristics among pores and cracks. The lamellar organic-rich felsic shale was chosen from K1qn1 formation at 2080.46 m in well X1. The mineral composition of shale is as follows: clay minerals (39.1%), quartz (37.8%), feldspar (10.4%), calcite (6.9%), siderite (0.4%), and pyrite (4.9%). The results of the organic geochemical experiments are as listed below: TOC (2.01 wt.%), $S_1$ (1.77 mg/g), $S_2$ (7.54 mg/g), $R_o$ (1.13%).

### 2.2. Ar-ion Milled FESEM

Sample surfaces were polished by using the Ar-ion milling (AIM) method in a GATAN Inc, PIPS (precision ion polishing system) [47]. Polished surfaces were oriented perpendicular to bedding. The treated samples not only had a smooth surface without damage, but also restored the real structure inside the rock, and clear mineral boundaries and micro/nano pores could be observed.

Field emission scanning electron microscopy (FE-SEM) used ZEISS GEMINI 500 field emission scanning electron microscopy [47]. The instrument had ultra-high resolution. It could observe and process secondary electron images and reflected electron images of various solid sample surface morphologies. Images for pore identification were taken at a working distance of 4 mm, voltage of 2.0 kV, contrast 70, brightness 55, magnification 1500×, resolution 8192 × 6144, and swelling time of 8 μs. In addition, 5 × 5 groups of images with the same magnification were captured to form the large field of high-resolution images.

### 2.3. Automatic Porosity Qquantification and Determination of REA

A method for automatic shale porosity quantification using an edge-threshold automatic processing (ETAP) technique was applied on all SEM images [29]. The first stage entailed converting the SEM image of the shale into an 8-bit greyscale image, counting the number of pixels in each greyscale range from 0–255, and drawing a relationship curve

between the number of pixels and greyscale. The relationship curve gave the grey cut-off values for the pore ($P_{cutoff}$) and the kerogen ($K_{cutoff}$). The initial pore image and the initial kerogen-pore image were then obtained by segmenting the original shale greyscale images using $P_{cutoff}$ and $K_{cutoff}$, respectively. The initial pore image and initial kerogen-pore image were superimposed, and the total of the peripheral length, area, and inner circumference of each isolated connected area in the kerogen-pore image was counted as $L_{ki}$; area $S_{ki}$, long-axis $L_{li}$, and short-axis $L_{si}$ were also counted. Additionally, $S_{pij}$ was calculated using the combined areas of all the pores in the superimposed initial pore. When determining whether an isolated connected area in the initial kerogen-pore image was a kerogen or mineral pore, the maximum area $S_{pijmax}$ of each isolated connected area in the kerogen-pore image was found. The kerogen region discriminant function $Q_{sti} = (S_{pijmax}/S_{ki})/[L_{ki}/S_{ki}/(L_{li}/L_{si})]$ was then used to determine whether the isolated connected area was a kerogen or mineral pore. The isolated connecting area in the original kerogen-pore image was kerogen when $Q_{sti}$ was less than or equal to 1, and it was an inorganic pore when $Q_{sti}$ was larger than 1. To determine the edge of the kerogen pores, the edge detection method was applied to the kerogen region image. An error function, $Q_{error}$, was set as follows: $Q_{error} = (A_{outside}+A_{inside})/A_{inside}$, where the value $Q_{error}$ showed the effectiveness of the analysis. Values were iteratively rebalanced to achieve the maximum $A_{inside}$ and, as a result, to reduce $Q_{error}$ to achieve the best performance. The discriminant function $Q_t$ was set to $Q_t = (A_{inside}/Q_{error})/A_{boundary}$, where Aboundary was the total pixels of pores; it was applied to normalize the value of $A_{inside}/Q_{error}$. When $Q_t$ reaches its maximum value, the optimal threshold was obtained. At this step, the pore image from the best threshold and the pore image from the edge detection approach were combined. Finally, $L_{li}/L_{si}$ was used as a discriminant parameter to characterize the pore and crack (pore: $L_{li}/L_{si} < 6$; crack: $L_{li}/L_{si} \geq 6$) based on the extracted organic pores and inorganic pores. Subsequently, we extracted five binarized images from the SEM images: region of OM (organic matter), Ops (organic pores), Ips (inorganic pores), OCs (organic cracks) and ICs (inorganic cracks).

The quantitative parameters for each pore were derived from ImageJ software (Figure 2). Then, the Kraver method [22] was used to calculate the circularity, convexity, and elongation.

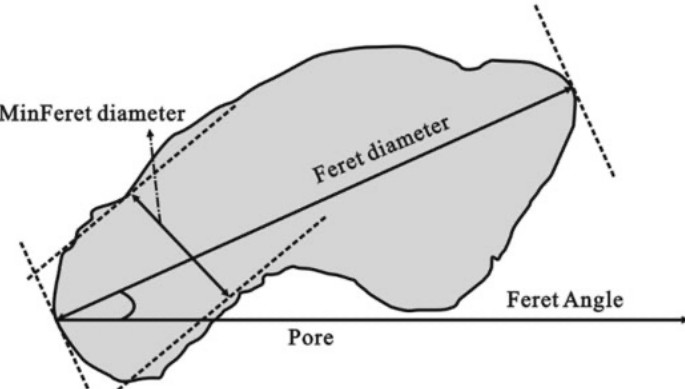

**Figure 2.** Illustration of the pore space parameters (Feret angle, Feret diameter, and MinFeret diameter) [48].

We used the box-counting method [48] to determine the representative elementary area (REA). Five parameters (the total surface porosity, surface porosity of Ops and OCs, surface porosity of Ips and ICs, surface porosity of OM, percentage of surface porosity of Ops and OCs/surface porosity of Ips and ICs) were calculated within increasing box sizes ranging from $1 \times 1$ to $76 \times 76$ in a single image.

### 2.4. Multifractal Method

The pore binarized image is considered as data set *I*. The data set *I* is segmented into $N(r)$ partition with scale $r$. $N(r)$ is equal to $2^n$, and $n$ is a positive integer.

$$N(r) = \frac{I}{r} = 2^n \tag{1}$$

The probability measure of the *i*-th partition set at scale $r$ can be expressed as equation

$$P_i = \frac{S_i(r)}{\sum_{i=1}^{N(r)} S_i(r)} \tag{2}$$

where $S_i(r)$ is the surface porosity of *i*-th partition.

When the dataset *I* has multiple fractal characteristics, the probability measure $P_i(r)$ and the scale $r$ are in a power exponential relationship.

$$P_i(r) \propto r^{a_i} \tag{3}$$

where $a_i$ is the coarse Holder index or singular intensity, which represents the density of the data distribution of the *i*-th partition set.

Therefore, different partition sets may have the same singular intensity. Using $N_a(r)$ to denote the cumulative proportion of the partition sets with singular intensity distribution between $a$ and $a \pm da$.

$$N_a(r) \propto r^{-f(a)} \tag{4}$$

where, $f(a)$ is the singular spectrum.

The partition function is defined as

$$Z(q, r) = \sum_{i=1}^{N(r)} p_i(r)^q \propto r^{\tau(q)} \tag{5}$$

where, $q$ is the weight factor, which ranges from $[-\infty, +\infty]$, $\tau(q)$ is the mass function.

$$\tau(q) = -\lim_{r \to 0} \frac{\log Z(q, r)}{\log r} = -\lim_{r \to 0} \frac{\log \sum_{i=1}^{N(r)} p_i(r)^q}{\log r} \tag{6}$$

We can define the generalized dimension $D(q)$ as Equation (7) using $P_i(r)$ and $q$.

$$D(q) = \begin{cases} \frac{\tau(q)}{1-q} = \frac{1}{1-q} \frac{\log \sum_{i=1}^{N(r)} p_i(r)^q}{\log r}, q \neq 1 \\ \frac{\sum_{i=1}^{N(r)} p_i(r) \log p_i(r)}{\log r}, q = 1 \end{cases} \tag{7}$$

The relationship between $\alpha(q)$, $\tau(q)$ and $f(a)$ can be obtained from the Legendre transformation.

$$\alpha(q) = \frac{d\tau(q)}{dq} \tag{8}$$

$$f(a) = q\alpha(q) - \tau(q) \tag{9}$$

## 3. Results and Discussion

### 3.1. REA Determination

The pore characterization reliability depends on the appropriate REA. In general, the pore parameters vary as the box area increases. When the box area is larger than the REA, the pore parameters do not vary significantly. The REA of shale has been studied based on the point counting method, with values around 0.01 mm² [48–50]. However, these studies focus on the surface porosity representability and ignore the pore type. In this study, we considered both the pore parameter and pore type to determine the REA. Figure 3a shows

the different sizes of boxes selected for determining the REA, with values varying from $1 \times 1$ to $76 \times 76$. When the REA is greater than $15 \times 15$, total surface porosity becomes stable, yet surface porosity of OM remains unstable. However, the surface porosity no longer varies for all pore types when the REA exceeds $35 \times 35$ (Figure 3b). At this point, the REA can represent the pore type and porosity characteristics of the shale.

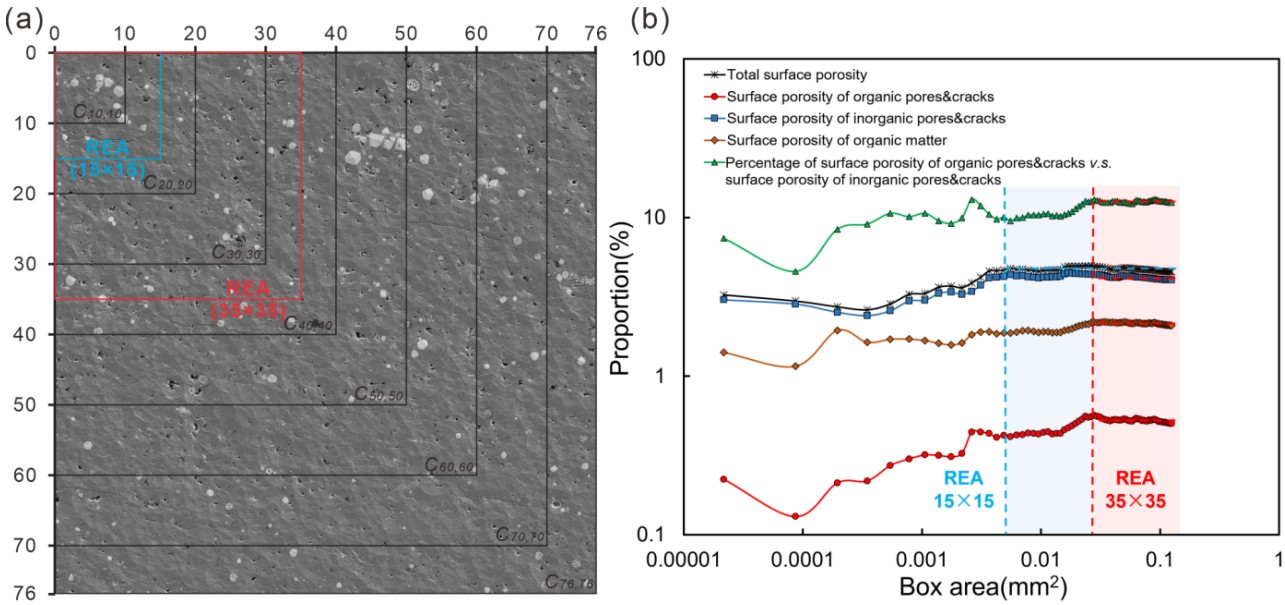

**Figure 3.** (**a**) Principle of REA determination, blue box is regular REA, red box is stringent REA; (**b**) the variation of surface porosity for the four pore types versus box area.

### 3.2. Porosity from FE-SEM

According to the relationship between pores and mineral and organic matrix particles, shale reservoir space can be divided into four categories: OPs, IPs, OPs, ICs (Figure 4a).

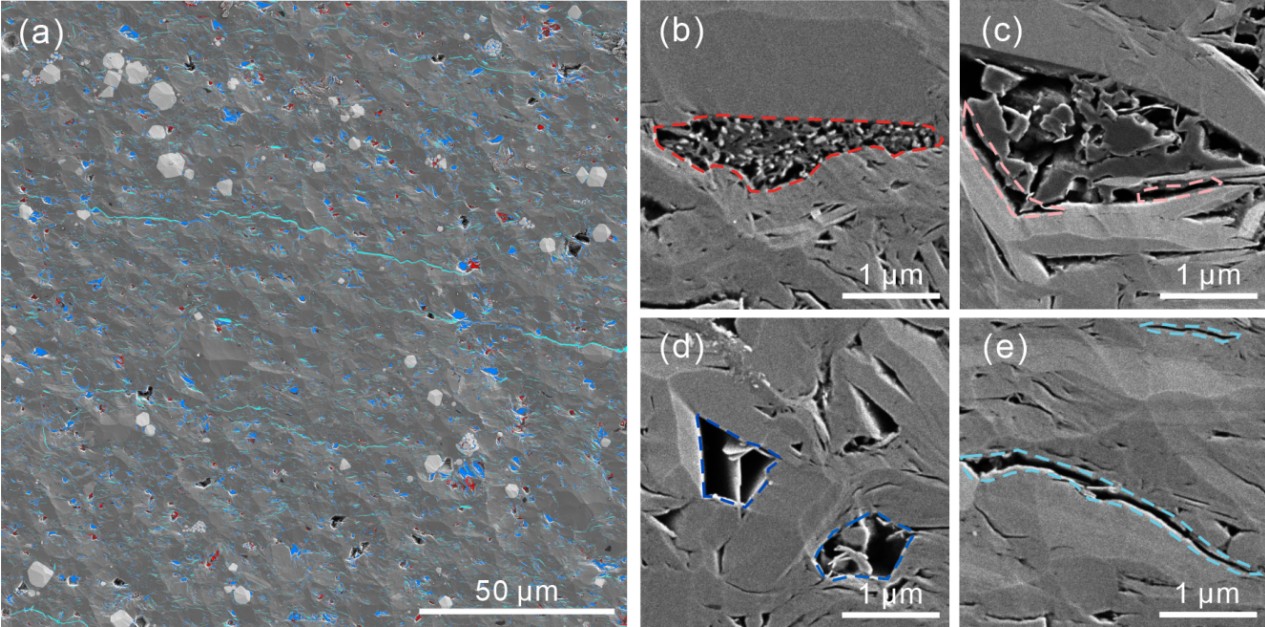

**Figure 4.** (**a**) The result of REA with resolution of 9.3 nm, (**b**) organic pores; (**c**) organic cracks, (**d**) inorganic pores, (**e**) inorganic cracks. (Red areas are organic pores, pink areas are organic cracks, deep blue areas are inorganic pores, light blue areas are inorganic cracks.)

A total of 35,948 OPs were identified for this study, which is about 11.59% of the total visible surface porosity (Table 1). This reveals that OPs are not the main pore type of the Qingshankou Shale. OPs are the minimum in size and possess the smallest pore area and radius (Table 1 and Figure 4b).

**Table 1.** Pore characteristic results for four pore types.

| Pore Type | Pore Area Range (nm$^2$) | Perimeter Range (nm) | Feret Range (nm) | MinFeret Range (nm) | Area (%) | Quantity |
|---|---|---|---|---|---|---|
| Organic pores | 346 [1]–1,371,731 [2] 2906 [3] | 53–18,944 193 | 26–3082 73 | 9–1610 38 | 11.59 | 35,948 |
| Inorganic pores | 346–6,525,671 22,951 | 53–65,881 780 | 26–12,983 275 | 9–3684 101 | 63.56 | 36,248 |
| Organic cracks | 519–407,195 15,874 | 79–4961 603 | 34–2315 282 | 9–237 51 | 0.61 | 498 |
| Inorganic cracks | 865–5,250,808 24,155 | 188–70,587 1321 | 93–22,088 562 | 9–2512 81 | 24.24 | 13,453 |

Note: Superscripts 1, 2, and 3 indicate the minimum, maximum and average values, respectively.

IPs (Table 1 and Figure 4d) are mainly intergranular pores of clay minerals and intergranular pores of quartz particles. IPs account for 63.56% of the overall pore space and are an important part of the Qingshankou Shale (36,248 pores were identified in this time). The pore area of IPs varies from 346 to 6,525,761 nm$^2$ (mean 22,951 nm$^2$), and the range of perimeter is from 53 to 65,881 nm (mean 780 nm). The average values of Feret diameter and MinFeret diameter are 275 and 101, respectively.

The development of OCs in the shale is the lowest, accounting for only 0.61% of the total pore space (Table 1). It can be seen that OCs at the organic particle boundary are parallel to the edge (Figure 4c). The Feret diameter of OCs is small with a range of 34–2315 nm, which means that the pores cannot be connected by organic crack effectively.

ICs are the most widely distributed fractures in the shale matrix. A total of 13,453 ICs was detected, contributing 24.24% to the total pore space (Table 1). It is demonstrated that ICs are distributed between mineral grains with surprising extension length; the Feret diameter is up to 22088 nm (Figure 4e). It can be concluded that ICs are an important factor in the flowability of shale oil.

Four morphological parameters were calculated according to the Klaver method [22] for characterizing the morphological information of the pores: circularity, convexity, elongation, and Feret angle. Circularity values range between 0 and 1, where a circularity value of 1 represents a perfect circle. Convexity (0–1) indicates the roughness of the pore surface, a value of 0 means absolutely smooth. The elongation (0–1) represents the relationship between length and width of the pores, the closer to 1, the narrower and longer the pores are [22].

The distribution of OPs circularity is obviously skewed to the right, with a frequency peak of between 0.7 and 1.0, which implies that OPs are closest to a circle (Figure 5a). IPs have poor circularity with a frequency peak of between 0.4 and 0.5, reflecting the irregular shape of the particles constituting the IPs. The cracks are supposed to have a minimum circularity. However, the OCs have a higher circularity with a frequency peak of between 0.3 and 0.5. It is demonstrated that the pores have a smoother surface than the cracks, and the OPs have the smoothest surface (convexity up to 0.9) (Figure 5b). It is indicated that the ICs and IPs elongation (peak at 0.9/0.7) is greater than OCs and OPs elongation (peak at 0.8/0.5) (Figure 5c), which is consistent with observation in the SEM images. Finally, the directionality (Feret Angle) of the four pore types is discussed (Figure 5d). The curves for the four pore types show a similar two-peak pattern, appearing at 40° and 160°.

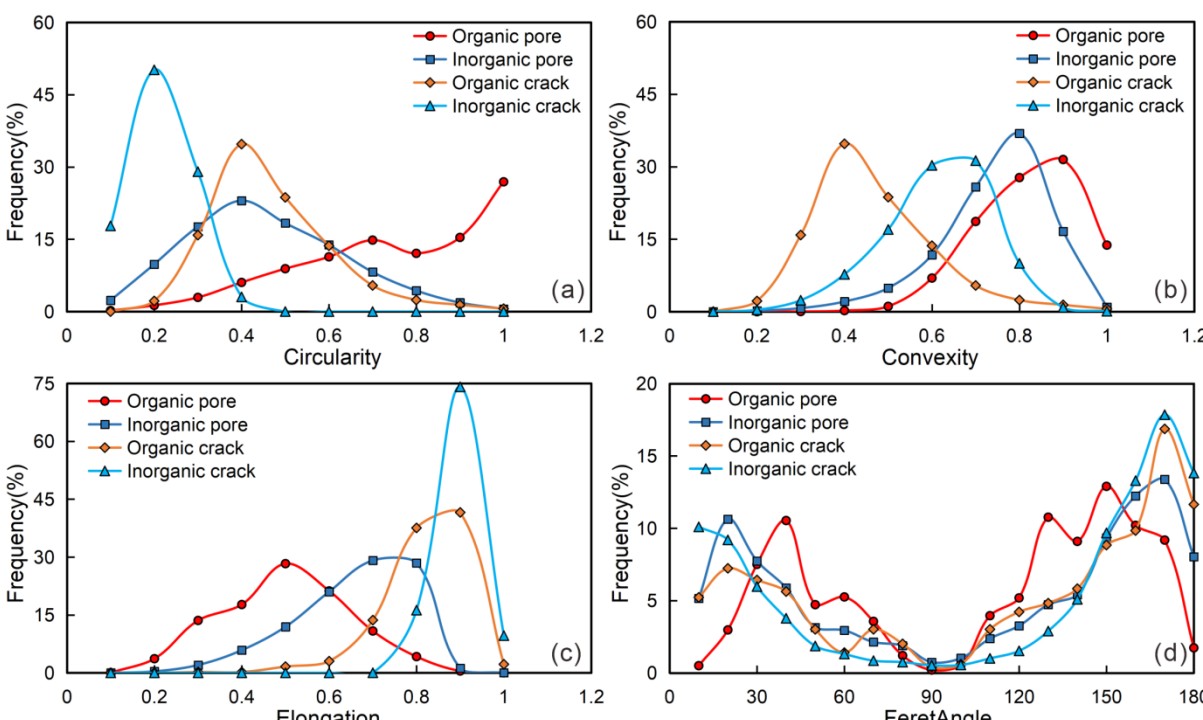

**Figure 5.** Frequency plots of (**a**) pore circularity, (**b**) pore convexity, (**c**) pore elongation, and (**d**) pore FeretAngle.

The seepage capacity of shale oil is controlled by the diameter of the pore along the short axis (MinFeret diameter), thus the normalization distributions of MinFeret diameter was calculated (Figure 6). The results show that micropores (100–1000 nm) are the main component of the Qingshankou shale, occupying 69% of the total pore. Of the four pore types, the IPs had the largest proportion of micropores (71.19%), followed by ICs (63.14%), OPs (55.14%) and OCs (51.52%).

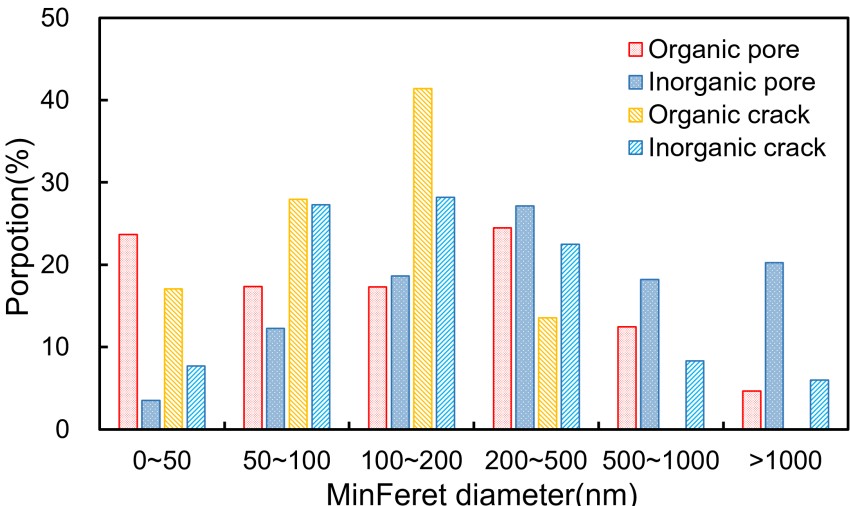

**Figure 6.** Pore size distributions of four pore types.

### 3.3. Multifractal Characteristics

The multifractal characteristics of the four pore types are calculated and discussed in this section. The q is set from −20 to 20 with an interval of 0.5. The distribution curves of q~Dq and α~f(α) for different pore types are shown in Figure 7.

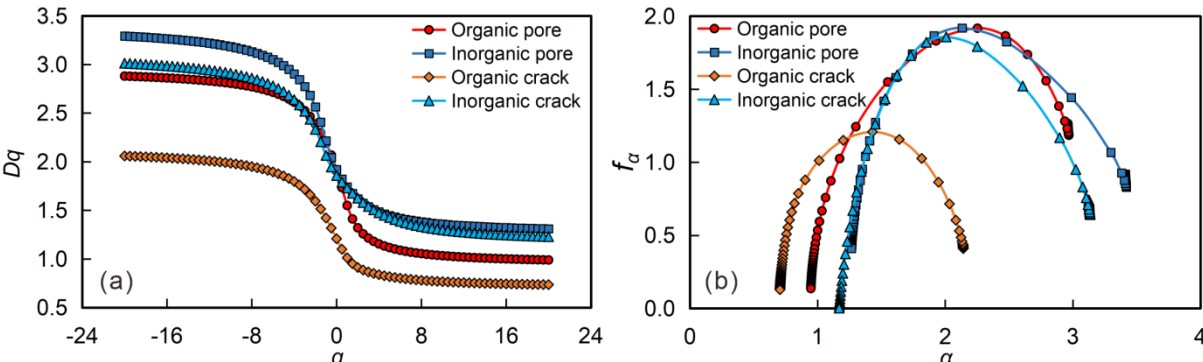

**Figure 7.** (**a**) Generalized dimensional and (**b**) multifractal spectra of four different pore types.

Dq decreases rapidly with increasing q when q < 0, and the Dq varies greatly among pore type. Dq decreases slowly with the increase of q when q > 0, and the difference of Dq between pore type was small (Figure 7a). A multitude of pore structure parameters $(D_0, D_1, D_2)$ can be obtained by Dq-q. $D_0$ refers to the capacity dimension characterizing the average pore structure distribution and reflecting the pore structure complexity [48]. The information dimension and the correlation dimension are denoted as $D_1$ and $D_2$, respectively. $D_1/D_0$ indicates the dispersion of the porosity to the pore diameter [33] (Table 2).

**Table 2.** Multifractal parameters of four pore types.

| Pore Type | $D_0$ | $D_1$ | $D_2$ | $D_1/D_2$ | $\alpha_{min}$ | $\alpha_{max}$ | $\alpha_0$ | $\Delta\alpha$ | A |
|---|---|---|---|---|---|---|---|---|---|
| Organic pore | 1.92 | 1.55 | 1.32 | 0.81 | 0.95 | 2.97 | 2.25 | 2.02 | 1.82 |
| Inorganic pore | 1.21 | 1.01 | 0.91 | 0.84 | 0.71 | 2.14 | 1.43 | 1.44 | 1.02 |
| Organic crack | 1.92 | 1.74 | 1.62 | 0.91 | 1.27 | 3.42 | 2.13 | 2.16 | 0.67 |
| Inorganic crack | 1.86 | 1.73 | 1.63 | 0.93 | 1.16 | 3.14 | 2.01 | 1.97 | 0.75 |

Each pore type has typical multiple fractal characteristics, exhibiting $D_0 > D_1 > D_2$. Ops and OCs have the largest $D_0$ values (1.918/1.919), while IPs have the smallest $D_0$ values (1.208) (Table 2). It can be concluded that Ops and OCs have the most complex structure, while OPs are simpler. The distribution of $D_1$ and $D_2$ for each pore type was the same as each other. OCs and ICs have higher $D_1/D_0$ value (0.91/0.93), thus the pore volume varies greatly at different scales; whereas the pore volume of the Ops and IPs are more concentrated at different scales with lower $D_1/D_0$ value (0.81/0.84) (Table 2).

The multifractal spectral singularity index can effectively scale the pore structure heterogeneity, $\alpha_{max}$ and $\alpha_{min}$ indicate the maximum and minimum values of multifractal spectral singularity index, respectively (Figure 7b). $\Delta\alpha$ is the difference between $\alpha_{max}$ and $\alpha_{min}$, namely $\Delta\alpha = \alpha_{max} - \alpha_{min}$. $\alpha_0$ indicates the singularity index when q = 0. Multifractal spectral skewness $A = (\alpha_0 - \alpha_{min})/(\alpha_{max} - \alpha_0)$. $\Delta\alpha$ and $\alpha_0$ indicate the non-homogeneity of the pore structure. The larger the values are, the greater the heterogeneity of the pore structure is [48] (Table 2).

The distribution of $\Delta\alpha$ and $\alpha_0$ is consistent. The $\Delta\alpha$ varies from 1.44 to 2.02 ($\alpha_0$ between 1.43 and 2.25), which is significantly lower than the value of Shahejie shales in the Bohaiwan basin, which is the most important shale oil producer in China [48]. It can be concluded that the pore heterogeneity of Qingshankou Shale is weaker. OPs ($\Delta\alpha = 2.02$, $\alpha_0 = 2.25$) are more heterogeneous than IPs ($\Delta\alpha = 1.44$, $\alpha_0 = 1.43$), and cracks (average $\alpha_0 = 2.07$, average $\Delta\alpha = 2.06$) are more heterogeneous than pores (average $\alpha_0 = 1.84$, average $\Delta\alpha = 1.73$). This conclusion is consistent with the result from Dq-q. In addition, the As of the pores (average A = 0.71) are all greater than those of the cracks (average A = 0.92) which indicates that the pores have a lower index and fluctuations (Table 2).

## 4. Conclusions

The stringent REA selection requires that the porosity of each pore type does not change significantly with box size. The stringent REA of high-resolution SEM image was identified to be $35 \times 35$ for lacustrine oil-prone shale from the Qingshankou Formation. A stringent REA was chosen to analyse the pore structure. The multifractal theory was applied to explain the heterogeneity of each pores-type network.

Four pore types were found in the stringent REA: organic pores, organic cracks, inorganic pores, inorganic cracks. Inorganic pores and inorganic cracks were the main pore types and accounted for 87.8% of the total pore volume, and organic cracks were the least important to the Qingshankou shale.

Organic pores are the most complex for pore structure, the least average MinFeret diameter (38 nm), but the largest $D_0$, $\alpha_0$, A (1.92, 2.25, 1.82). Moreover, the extension distance of the organic cracks is short and cannot effectively connect the organic pore.

Inorganic pores are the simplest for pore morphologies, the maximum average MinFeret diameter (101 nm), and the minimum $D_0$, $D_1$, $D_2$, $D_1/D_2$, $\alpha_0$, $\Delta\alpha$ (1.21, 1.01, 0.91, 0.84, 1.43, 1.44). In addition, the inorganic cracks have long extension distances and stronger homogeneity, which can effectively connect the inorganic pores.

Therefore, we conclude that inorganic pores and inorganic cracks are a key factor in the storage and seepage capacity of Qingshankou shale. Organic pore and organic cracks provide limited storage space.

**Author Contributions:** Conceptualization, S.T. and Z.D.; methodology, S.T.; software, Y.G.; validation, Y.G.; formal analysis, Z.D.; investigation, S.T.; resources, S.T.; data curation, S.T.; writing—original draft preparation, S.T.; writing—review and editing, Z.D.; visualization, Z.L.; supervision, Z.D.; project administration, S.T.; funding acquisition, S.T. All authors have read and agreed to the published version of the manuscript.

**Funding:** This study was partly funded by China Postdoctoral Science Foundation (2020M670878), Superior Youth Foundation of Heilongjiang Province (YQ2020D002), Project of Daqing Guiding Science and Technology Plan (zd-2019-18), Science and Technology Project of Heilongjiang Province (No.2020ZX05A01).

**Institutional Review Board Statement:** Not applicable.

**Data Availability Statement:** Not applicable.

**Conflicts of Interest:** We declare that we have no financial or personal relationships with other people or organizations that can inappropriately influence our work, there is no professional or other personal interest of any nature or kind in any product, service and/or company that could be construed as influencing the position presented in, or the review of, the manuscript entitled, "Pore microstructure and multifractal characterization of lacustrine oil-prone shale using high-resolution SEM: A case sample from natural Qingshankou shale".

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
