# Peer review of "Pore Microstructure and Multifractal Characterization of Lacustrine Oil-Prone Shale Using High-Resolution SEM: A Case Sample from Natural Qingshankou Shale"

_fractalfract, doi:10.3390/fractalfract6110675_

Round 1
Reviewer 1 Report
The work is devoted to the experimental study of pore microstructure and their multifractal characteristic for a sample of lacustrine oil shale from Qingshankou. The work contains new material and is of some applied interest for reservoir evaluation, capacity control and hydrocarbon transportation properties. The work can be recommended for publication after eliminating the following comments.
1. In the work rather superficially described the methods of analysis and research used. For example, line 104 mentions "The results of the organic geochemical experiments ...", but how and for what these experiments were carried out is not specified.
2. The problem of delineation of certain areas (in this case, pores) from the results of image analysis is a non-trivial problem, there are many different methods of solving this problem. In the paper, line 132 says "When determining whether an isolated connected area...", in line 139 "To determine the edge of the kerogen pores, the edge detection method was applied...". However, the article does not specify what methods of identification and analysis of pore boundaries were used by the authors. This issue should be described in more detail in terms of describing the methodology, since this issue is the key to this study.
3. Line 164 says "The pore binarized image is considered...". How were these binarized images obtained from the original images, what was the threshold of sensitivity of transformation used? The final result depends on this.
4. It is not clear from the description how the authors distinguish different types of pores (organic pores, organic cracks, inorganic pores, inorganic cracks) from the image analysis data. This issue should be stated in more detail.
5. There are a number of unfortunate expressions in the work, for example on line 110 "...micro nano pores...", on line 112 "Field emission scanning electron microscopy (FE-SEM) used ZEISS GEMINI 500 field emission scanning electron microscopy" and others.
Author Response
The work is devoted to the experimental study of pore microstructure and their multifractal characteristic for a sample of lacustrine oil shale from Qingshankou. The work contains new material and is of some applied interest for reservoir evaluation, capacity control and hydrocarbon transportation properties. The work can be recommended for publication after eliminating the following comments.
- In the work rather superficially described the methods of analysis and research used. For example, line 104 mentions "The results of the organic geochemical experiments ...", but how and for what these experiments were carried out is not specified.
Response:Thank you. We have added the procedure for the sample geochemical experimental data [1-4]. The pore space of shale is closely related to its material composition. Therefore, we need to understand the mineral composition and organic matter characteristics of shale.
Reference:
- Barker, C. E.; Pawlewicz, M., The correlation of vitrinite reflectance with maximum temperature in humic organic matter. Paleogeothermics 1986, 79-93.
- Whittig, L.; Allardice, W., X‐ray diffraction techniques. Methods of Soil Analysis: Part 1 Physical and Mineralogical Methods 1986, 5, 331-362.
- Behar, F.; Beaumont, V.; Penteado, H. D. B., Rock-Eval 6 technology: performances and developments. Oil & Gas Science and Technology 2001, 56, (2), 111-134.
- Jarvie, D. M., Total organic carbon (TOC) analysis: Chapter 11: Geochemical methods and exploration. 1991.
- The problem of delineation of certain areas (in this case, pores) from the results of image analysis is a non-trivial problem, there are many different methods of solving this problem. In the paper, line 132 says "When determining whether an isolated connected area...", in line 139 "To determine the edge of the kerogen pores, the edge detection method was applied...". However, the article does not specify what methods of identification and analysis of pore boundaries were used by the authors. This issue should be described in more detail in terms of describing the methodology, since this issue is the key to this study.
Response:Thank you for the kind remind. We now provide a more detailed description of pore identification (Figure 1). Firstly, the inorganic pore and the kerogen region were distinguished according to a kerogen discriminant function (eqv. 1). To determine the edge of the kerogen pores, the edge detection method was applied to the kerogen region image. The Sobel [5], Prewitt[6], Roberts [7] and Canny [8] operators were used to extract the edges of the original shale SEM grayscale images, and the edges obtained by the four operators were merged, and the processed edge image was used as the calibration image. Then, the optimal threshold (45) of discrimination between the kerogen region and the organic pore was determined based on the constructed error function.
The kerogen region discriminant function Qsti was then used to determine whether the isolated connected area was a kerogen or mineral pore .
Qsti=(Spijmax/Ski)/[Lki/Ski/(Lli/Lsi)] (1)
The isolated connecting area in the original kerogen-pore image was kerogen when Qsti was less than or equal to 1, and it was an inorganic pore when Qsti was larger than 1.
The pixels which were located inside the region of the closed edge were counted as Ainside , the higher the value Ainside was, the more accurate the result should be. The pixels which were located outside the region of the closed edge were counted as Aoutside , and an error function was set as following:
Qerror = (Aoutside+Ainside)/Ainside (2)
where the value Qerror showed the effectiveness of the result. Values were iteratively rebalanced to achieve the maximum Ainside and, as a result, to reduce Qerror to achieve the best performance. The discriminant function Qt was set as:
Qt = (Ainside/Qerror)/Aboundary (3)
Where Aboundary was the total pixels of pores. It was applied to normalize the value of Ainside/Qerror. When Qt reached its maximum value, the optimal threshold was obtained.
Figure 1. Technical roadmap for the edge-threshold automatic processing (ETAP) method for pore identification and quantification in SEM images [9].
Reference:
- Kanopoulos, N.; Vasanthavada, N.; Baker, R. L., Design of an image edge detection filter using the Sobel operator. IEEE Journal of solid-state circuits 1988, 23, (2), 358-367.
- Yang, L.; Wu, X.; Zhao, D.; Li, H.; Zhai, J. In An improved Prewitt algorithm for edge detection based on noised image, 2011 4th International congress on image and signal processing, 2011; IEEE: pp 1197-1200.
- Muthukrishnan, R.; Radha, M., Edge detection techniques for image segmentation. International Journal of Computer Science & Information Technology 2011, 3, (6), 259.
- Rong, W.; Li, Z.; Zhang, W.; Sun, L. In An improved CANNY edge detection algorithm, 2014 IEEE international conference on mechatronics and automation, 2014; IEEE: pp 577-582.
- Tian, S.; Brown, L.; Zeng, F.; Xue, H.; Lu, S.; Erastova, V.; Greenwell, C. In A Method for Automatic Shale Porosity Quantification Using New Advanced Edge Threshold Technique, Geophysical Research Abstracts, 2019.
- Line 164 says "The pore binarized image is considered...". How were these binarized images obtained from the original images, what was the threshold of sensitivity of transformation used? The final result depends on this.
Response:Thank you. The initial pore image and the initial kerogen-pore image were then obtained by segmenting the original shale greyscale images using Pcutoff (32) and Kcutoff (70), respectively. Note that the final results are subjected to kerogen region discrimination and organic pore discrimination for the segmented images.
- It is not clear from the description how the authors distinguish different types of pores (organic pores, organic cracks, inorganic pores, inorganic cracks) from the image analysis data. This issue should be stated in more detail.
Response:Thank you. We have added table 1 for classifying the different pore types.
Table1. Pore type classification scheme.
|
Pore type |
Qsti |
Lli/Lsi |
|
Organic pores |
≥1 |
<6 |
|
Inorganic pores |
<1 |
<6 |
|
Organic cracks |
≥1 |
≥6 |
|
Inorganic cracks |
<1 |
≥6 |
Note: Qsti kerogen region discriminant function, long-axis distance Lli, and short-axis Lsi distance
- There are a number of unfortunate expressions in the work, for example on line 110 "...micro nano pores...", on line 112 "Field emission scanning electron microscopy (FE-SEM) used ZEISS GEMINI 500 field emission scanning electron microscopy" and others.
Response:We apologize for the inappropriate description. They have been corrected

Reviewer 2 Report
In this manuscript, a natural lacustrine oil-prone shale in the Qingshankou Formation of Songliao Basin is used as the research object. Based on the FE-SEM, high-resolution cross-section of shale were obtained to analyze the microstructure of pores and characterize the heterogeneity of pores by multifractal theory. The stringent representative elementary area (REA) of SEM crosssection was determined to be 35×35. Four pore types were found and analyzed in the stringent REA: organic pores, organic cracks, inorganic pores, inorganic cracks. The results showed that inorganic pores&cracks were the main pore types, and organic cracks were the least importance to the Qingshankou shale. Inorganic pores were characterized as the simplest pore morphologies, and organic pores were found to be the most complex for pore structure.
The heterogeneity of pores in Qingshankou shale is studied quantitatively and some conclusions are obtained, but there are still some problems to be pointed out.
- An important issue is that there is no reference to experimental groups in the body of the paper. The authors need to elaborate on the overall situation that was analyzed and processed.
- There does not appear to be a drawing of error bars in all statistical plots in the text, and is this due to the number of experimental groups having only one? If there is only a single group of samples, can this group be representative of the overall situation of Qingshankou shale?
- How are the three parameters D0, D1, D2 mentioned in line 262 obtained by Dq-q?
- Should the "while OPs are simple" in line 270 be changed to " while IPS are simple "? Is it a clerical mistake?
- The important application of the pores of Qingshankou shale studied in this paper is for shale oil storage. The variety and heterogeneity of pores in Qingshankou shale were obtained in this paper using various morphological parameters and calculation methods to characterize pores, but how do these parameters affect the storage of Qingshankou shale for shale oil?
- This paper is more about the calculation and analysis of parameters, but can these guide subsequent applications? Please expand instructions in the article.
- This paper classifies the pores types of Qingshankou shale into four categories. What is the difference between organic and inorganic pores? Are the differences in morphological parameters between them due to differences in their structure and composition? If not, what accounts for these differences?
- The research width of this paper is good, but the research depth is not enough, and the content is not very rich.
Author Response
Reviewer 2:
In this manuscript, a natural lacustrine oil-prone shale in the Qingshankou Formation of Songliao Basin is used as the research object. Based on the FE-SEM, high-resolution cross-section of shale were obtained to analyze the microstructure of pores and characterize the heterogeneity of pores by multifractal theory. The stringent representative elementary area (REA) of SEM crosssection was determined to be 35×35. Four pore types were found and analyzed in the stringent REA: organic pores, organic cracks, inorganic pores, inorganic cracks. The results showed that inorganic pores&cracks were the main pore types, and organic cracks were the least importance to the Qingshankou shale. Inorganic pores were characterized as the simplest pore morphologies, and organic pores were found to be the most complex for pore structure.
The heterogeneity of pores in Qingshankou shale is studied quantitatively and some conclusions are obtained, but there are still some problems to be pointed out.
1.An important issue is that there is no reference to experimental groups in the body of the paper.
The authors need to elaborate on the overall situation that was analyzed and processed.
Response:Thank you. The focus of this article is to introduce a method for characterizing pore types. Therefore, there is no experimental group designed. In the future work, we will consider the experimental and control groups to analyze the overall characteristics of pore space in the Qingshankou Shale.
2.There does not appear to be a drawing of error bars in all statistical plots in the text, and is this due to the number of experimental groups having only one? If there is only a single group of samples, can this group be representative of the overall situation of Qingshankou shale?
Response:Thank you. This sample may not represent the overall characteristics of Qingshankou shale. We are more concerned with the method of pore extraction and fractal algorithm. In the next stage, we will design a more detailed sample list to analyze the overall characteristics of the Qingshankou Shale.
3.How are the three parameters D0, D1, D2 mentioned in line 262 obtained by Dq-q?
Response:Thank you. In Figure 2(a), D0, D1, D2 can be obtained at q=0,1,2.
Figure2. (a) Generalized dimensional and (b) multifractal spectra of four different pore types.
4.Should the "while OPs are simple" in line 270 be changed to " while IPS are simple "? Is it a clerical mistake?
Response:Sorry for my carelessness, we have made change.
5.The important application of the pores of Qingshankou shale studied in this paper is for shale oil storage. The variety and heterogeneity of pores in Qingshankou shale were obtained in this paper using various morphological parameters and calculation methods to characterize pores, but how do these parameters affect the storage of Qingshankou shale for shale oil?
This paper is more about the calculation and analysis of parameters, but can these guide subsequent applications? Please expand instructions in the article.
Response:Thank you for your interest, we will work on the application of research results in the future.
6.This paper classifies the pore types of Qingshankou shale into four categories. What is the difference between organic and inorganic pores? Are the differences in morphological parameters between them due to differences in their structure and composition? If not, what accounts for these differences?
Response:Thank you for the kind reminder. In this sample, the inorganic pores commonly exist in the interface between clastic grains. The clastic grains receive sufficient sorting and rounding after a long period of transport, and the shape is more regular. Therefore, inorganic pores have a relatively simple pore structure. The organic pores include the pores formed due to gas generated from liquid hydrocarbon pyrolysis, and the remaining pores formed due to hydrocarbon generation. The organic matter is irregular in shape and has heterogeneous thermal stability. Therefore, the pores of organic matter after pyrolysis are generally irregular.
The research width of this paper is good, but the research depth is not enough, and the content is not very rich.

Round 2
Reviewer 2 Report
This paper has been revised well and is acceptable for publication.